# Automatically Determining Lumbar Load during Physically Demanding Work: A Validation Study

**DOI:** 10.3390/s21072476

**Published:** 2021-04-02

**Authors:** Charlotte Christina Roossien, Christian Theodoor Maria Baten, Mitchel Willem Pieter van der Waard, Michiel Felix Reneman, Gijsbertus Jacob Verkerke

**Affiliations:** 1Department of Rehabilitation Medicine, University Medical Center Groningen, University of Groningen, Hanzeplein 1, 9713 GZ Groningen, The Netherlands; m.f.reneman@umcg.nl (M.F.R.); g.j.verkerke@gmail.com (G.J.V.); 2Department Rehabilitation Technology, Roessingh Research and Development, Roessinghsbleekweg 33b, 7522 AH Enschede, The Netherlands; c.baten@rrd.nl (C.T.M.B.); M.vanderWaard@rrd.nl (M.W.P.v.d.W.); 3Department of Biomechanical Engineering, University of Twente, Drienerlolaan 5, 7522 NB Enschede, The Netherlands

**Keywords:** physically active workers, low back pain, inertial motion units

## Abstract

A sensor-based system using inertial magnetic measurement units and surface electromyography is suitable for objectively and automatically monitoring the lumbar load during physically demanding work. The validity and usability of this system in the uncontrolled real-life working environment of physically active workers are still unknown. The objective of this study was to test the discriminant validity of an artificial neural network-based method for load assessment during actual work. Nine physically active workers performed work-related tasks while wearing the sensor system. The main measure representing lumbar load was the net moment around the L5/S1 intervertebral body, estimated using a method that was based on artificial neural network and perceived workload. The mean differences (MDs) were tested using a paired *t*-test. During heavy tasks, the net moment (MD = 64.3 ± 13.5%, *p* = 0.028) and the perceived workload (MD = 5.1 ± 2.1, *p* < 0.001) observed were significantly higher than during the light tasks. The lumbar load had significantly higher variances during the dynamic tasks (MD = 33.5 ± 36.8%, *p* = 0.026) and the perceived workload was significantly higher (MD = 2.2 ± 1.5, *p* = 0.002) than during static tasks. It was concluded that the validity of this sensor-based system was supported because the differences in the lumbar load were consistent with the perceived intensity levels and character of the work tasks.

## 1. Introduction

Physically active workers sometimes can experience muscle and spinal overload while performing their physically demanding jobs [1]. Such an overload is hypothesized to be due to a misbalance between the physical workload and the individual capacity of each worker [2]. This misbalance may cause health problems among these workers, such as musculoskeletal disorders like lower back pain [3,4,5,6,7]. These problems usually result in the loss of productivity [8,9,10], loss of quality and safety [1,11], and absenteeism [1,12]. Hence, to help prevent these health problems, to improve rehabilitation, and promote return to work and sustainable employability, it is important to investigate and optimize the musculoskeletal load while performing physically demanding jobs [13,14,15].

There is a need for a device that can measure the individual work-related lumbar load exposure objectively while performing a physically demanding job [16,17]. Typically, this lumbar load is represented by the net moment around the center of the intervertebral body at the spinal level L5/S1. Various methods have been developed to estimate the net moment in the lower back under known load-handling conditions [18,19,20,21,22]. All of these methods use 3D body segment kinematics data acquired using a marker-based motion analysis system, and load conditions were “known” through the direct measurement of the reaction forces exerted to the feet or hands or from detailed information on the loads handled.

Inertial motion capture systems seem useful for assessing work-related back load exposure in workers in industrial environments through monitoring the working postures of individuals and through driving biomechanical-linked segment models to estimate 3D net moments and forces in the lower spine [18,19,23,24,25,26,27,28,29,30]. Using inertial magnetic measurement units (IMMUs) allows for more freedom in 3D kinematics assessments in comparison to marker-based motion analysis systems [31]. However, the need to measure all the forces exerted on the human’s body, e.g., by force plates embedded in the lab floor, severely limits their practical applications. A more mobile alternative is to use instrumented shoes that measure ground reaction forces while walking around [19,20,32,33]. This system provides more freedom of movement but requires that no other external forces be exerted to the lower body (e.g., by leaning against a table or supporting the load being handled). Additionally, the relatively large weight of these shoes and their current design make them less usable in practice. Therefore, alternative methods for estimating lumbar load assessment were developed, that do not require force assessment. In this study, an artificial neural network (ANN)-based method was applied to estimate 3D net moments (L5/S1) which was driven by electromyographic (EMG) and trunk kinematics data. It was trained in supervised mode during the initial part of each session with a limited set of calibration trials [18,19,21,31]. Target net moments for training were generated by direct estimation using a linked segment model (LSM-based method [24,29]) scaled by the height and weight of the subject. The LSM-based method was just driven by the kinematics data from IMMUs on the trunk and arms. The weight and inertial properties of the load handled during the calibration trials have to be known and their kinematics are derived from hand kinematics. In the actual trials, a trained ANN-based method was used to estimate the net moments from the EMG magnitude data and the IMMU kinematics data and of a subject during actual work. This ANN-based method has been developed over several iterations from 1993 until the current date and has been evaluated in several studies, e.g., in direct comparison against a state-of-the-art laboratory-based method driven by marker-based kinematics and force plates, the results of which were found to be promising [24]. The same group also successfully applied and validated the same approach for estimating shoulder joint load exposure estimation, confirming feasibility [34]. This ANN-based method, driven by trunk muscles EMG amplitude and IMMU kinematics, is therefore considered potentially useful for monitoring mechanical workload in the context of the worker’s postures and movements, while performing physically demanding jobs [19,21,22]. Ultimately, this may represent a tool that provides workers and ergonomists with instant feedback, which may contribute to preventing excess load exposure for a worker without back complaints. It might also be used for workers returning to work after an injury, and/or as part of their rehabilitation. However, the validity and usability of this system in the uncontrolled real-life working environment of physically active workers are still unknown.

The objective of this study was to test the discriminant validity of an ANN-based method for load assessment during actual work. The research questions were: (1) What is the base quality of this ANN-based method in estimating lumbar load when applied to the trial data of this study with known load handling? (2) Can this ANN-based method detect differences in load intensity and perceived workload during light and heavy tasks? (3) Can the system detect differences in load-variability during the static and dynamics tasks? (4) Can the system detect (a) symmetrical lumbar load difference around the anterior–posterior, mediolateral and longitudinal axes?

## 2. Materials and Methods

### 2.1. Subjects

A total of 23 subjects participated in this study, all of whom were physically active workers recruited through flyers distributed within selected companies. These selected companies were active in medical disinfection care, industrial chemical cleaning, and technical services. All subjects were informed about the study through an information letter and received a verbal explanation before the start of the study. The inclusion criterion was being a physically active worker aged between 18 and 67. The exclusion criteria were having any cardiovascular diseases; using pacemakers or other vital electronic devices; having high levels of pain or injuries in the back, shoulders, or upper extremities; or being at an advanced stage (around 20 weeks) of pregnancy. The Medical Ethics Committee of the University Medical Center Groningen, the Netherlands, issued a waiver for this study, stating that it did not involve medical research according to the Dutch law (M17.208063), and all subjects signed an informed consent form.

### 2.2. Study Design and Procedures

Each session with every subject comprised three phases: (1) trials for upper body segment calibration; (2) trials with known loads for supervised training and training quality validation; and (3) trials illustrating performance during a set of work-related tasks.

In phase 1, the subjects were asked to perform a set of movements to calibrate the orientation of the IMMUs relative to the body segments, while wearing the complete sensor set-up with IMMUs (and bipolar surface electromyography (sEMG) electrodes). The resulting segment calibration parameters were used to translate all sensor casing kinematics within a session to body segment and joint kinematics. The set contained 90° trunk bending, 45° trunk lateroflexion, 45° trunk rotation, 45° shoulder flexion, and 90° shoulder abduction. This was repeated five times and followed by three seconds of standing in a neutral anatomic position with the arms hanging next to the body with thumbs pointing forward.

In phase 2, the ANN-based method performance was tested against the LSM-based method by comparing the estimated moments by both methods in bending, flexion, and rotation movements while hand holding a 6 kg load (question 1). For this, the ANN was first trained with net moments directly estimated with the LSM-based method from similar tasks handling no load (“0 kg task”) and holding a load of a 10 kg trunk. Based on previous experience (unpublished sensitivity study) a simple feed-forward ANN was used with 1 hidden layer of 31 elements and sigmoid transfer functions with a fixed training criterium: “sum of RMSE Mx, RMSE My and RMSE Mz < 10 Nm”. This required about 10 s of computer time on a regular desktop PC for a typical input vector with about 6000 elements. The ANN was driven by EMG amplitudes from all 4 locations plus 3D kinematics only from the 2 IMMU sensors positioned on the sternum and sacrum: per segment, the 3D angular velocity and 3D linear acceleration signals were used. The rationale for not using the available 3D orientation data was that: (1) information of inclination is already present in the 3D accelerometer signal; and (2) in an industrial environment, a substantial risk exists of having large errors in IMMU orientation estimates by the disturbance of the natural earth magnetic field or permanent magnetization of the magnetometer in the IMMUs [35]. In preparation for phase 3, the ANN-based estimator was trained for each subject separately using all their respective bending, flexion and rotation tasks for all three weights (0, 6 and 10 kg).

In phase 3, questions 2, 3, and 4 about the discriminant validity of the ANN-based method were explored. The subjects performed job-specific work-related tasks for 5–10 min of different load intensity and variability. For all these tasks, net moment curves were estimated using the ANN-based network trained at the end of phase 2. Additionally, these tasks were ranked according to the checklist of physical workload [36]. Before the start of the measurements, the subjects received a questionnaire to identify the daily tasks and the frequency, duration and perceived the workload per task. This physical workload questionnaire of Peereboom and Lange [36] contained the following questions: What are tasks you perform on a daily basis at work? How often are these tasks performed on an average day? How heavy do you find these tasks? All subjects ranked tasks according to the load, starting with the heaviest task [36]. From the list of work-related tasks, four tasks were chosen, which may vary from one individual to another, which represented: (1) a light task with a low workload on the lower spine; (2) a heavy task with a high workload on the lower spine; (3) a static task (working with the lumbar region in the same posture); and (4) a dynamic (lifting) task in different (spinal) working postures [37,38]. To explore questions 2 and 4, the task with the highest workload was selected as a heavy task, and the task with the lowest workload was selected as a light task. Net moment data curve appearances were discussed with respect to the trial perceived workload. To explore questions 3 and 4, the criterion for the static task was that the lumbar region was held in the same posture or joint position for at least 4 s throughout the task with low variances in the lumbar posture when changing the posture (ISO standard 11226:2000) [37]. The criteria for the dynamic task were as follows: the task must be a lifting one and the lumbar spine should vary in posture. After every task, the subjects were asked to rate the perceived workload of the three tasks using Borg CR-10 rating scale, ranging from 0 to 10 (0 = not burdensome; 10 = extremely heavy) [39].

### 2.3. Materials

#### 2.3.1. Surface Electromyography Acquisition

All sEMG recordings were performed using a wearable sEMG instrument (Polybench Dipha; Inbiolab, Roden, the Netherlands). Bipolar electrodes (Covidien Kendall^™^ H124SG Ag/AgCl electrodes; Medtronic, Minneapolis, MN, USA) with an interelectrode distance of 2 cm (heart to heart) were placed bilaterally on the longissimus thoracis muscles at L1 (±3 cm horizontal from L1) and the iliocostalis lumborum muscles at L2–L3 (±6.5 cm horizontal from L2–L3), along with a reference electrode placed at the processus spinosus of C7 (Roy et al., 1995) (see Figure 1).

#### 2.3.2. Kinematics Acquisition and Net Moment Estimation

Six wired IMMUs (MVN Awinda; Xsens, Enschede, the Netherlands) were used to record 3D body segment kinematics. The IMMUs were placed on the sternum, upper and lower arms, and pelvis (sacrum) [19,22,40], as shown in Figure 1. The sample rate was 50 Hz. All IMMU data-acquisition procedures, as well as the translation of IMMU casing kinematics data to body segment and joint kinematics data, were performed with the FusionTools/XCM software suite (Roessingh Research and Development, Enschede, the Netherlands) [19,23] using the Xsens application programming interface (API 4.7). Using the same software suite, EMG data preparation (amplitude estimation by smoothed rectification, intrapolated resampling to 50 Hz) and synchronization with IMMU data were performed. The IMMU and sEMG data were synchronized by tapping on two sEMG electrodes and one IMMU electrode before the start of the measurement. This created a marker (peak) in the data which was used to synchronize both systems in a semi-automated procedure indicating both markers, manually using a cursor. Estimated synchronization errors < 0.05 s.

The LSM was driven by kinematics derived from all 6 IMMUs: 3D orientation (represented by quaternions); 3D angular velocity; and 3D linear acceleration (only used in controlled environment). The ANN was driven by EMG amplitudes plus 3D kinematics from only the 2 IMMU sensors positioned on sternum and sacrum: per segment the 3D angular velocity and 3D linear acceleration were signals used. All load exposure estimations were also performed using this software suite, with both LSM-based and ANN-based methods, as well as the calculation of all the descriptive statistics of the net moment curves and root-mean-square error (RMSE) values and correlation coefficients, comparing the target and estimated net moments.

### 2.4. Data Analysis

To test the correlation between the ANN-based method and the LSM-based method (question 1) in phase 2, the main evaluation comprised a comparison of ANN-based estimated and target net moment trajectories in 3D. The primary outcomes were the 3D curves of the estimated and target moment, as well as of the moment norm, and their RMSE curves of the moment magnitude (||M||, calculated through the net moment vector norm). The evaluation was performed for every rotation axis separately and for the net moment magnitude by means of the visual inspection of data plots and also by evaluating RMSE values between the estimated and target moments and the Pearson’s correlation coefficient (*r* with 0.1 < *r* < 0.5 indicating a weak correlation, 0.5 < *r* < 0.7 indicating a moderate correlation, 0.7 < *r* < 0.9 indicating a good correlation and > 0.9 indicating a very good correlation [41]) and its squared value (*r*^2^) representing the amount of explained variance in estimation. Movement in the mediolateral transverse (*y*) axis was named “trunk bending”, and movement in the anterior–posterior (*x*) axis was named “trunk lateroflexion” and the combined trunk twisting movement mainly in the longitudinal (*z*) axis was named “trunk rotation”. If a strong correlation was found between the ANN-based method and the LSM-based method, the ANN-based method was judged to be of an acceptable level.

In phase 3, the results of the questionnaire were categorized on the basis of the perceived workload, with 1 meaning a light task and 5 meaning a very heavy task [36]. According to these scores, the tasks for test questions 2, 3 and 4 were selected. To test whether this sensor system could distinguish differences between the intensity and variability of the estimated lumbar load, a discriminant validity analysis was performed. The primary parameters that we compared were the mean, peak (max) and variance (standard deviation within a subject) of the net moment in the lumbar region and results. They were also discussed in relation to the perceived workload (Borg CR-10) [39].

To explore if this ANN-based method can distinguish the estimated lumbar load differences in intensity levels (question 2), differences between the light and heavy tasks were analyzed. The hypothesis was that during heavy tasks, the mean net moments in the lumbar region would be significantly higher. Additionally, the hypothesis that the perceived workload (Borg CR-10 score) of the heavy tasks was significantly higher than the light tasks was tested. The hypothesis of question 3 (variability level) was that the variance in the net moments in the lumbar region was higher during dynamic tasks than during the static tasks.

To explore the (a) symmetrical character of working postures, the movement direction of the moment around the anterior–posterior, mediolateral and longitudinal axes (question 4) was assessed. The net moment data were divided into two segments based on the direction of movement (positive or negative movement direction); anterior–posterior with lateroflexion to the left (positive) and right (negative), mediolateral with flexion (positive) and extension (negative), longitudinal with rotation to the left (positive) and right (negative) axes. This was done based on the hypothesis that the tasks had an asymmetrical character. These positive and negative moment segments were tested separately with the same hypothesis described for questions 2 and 3.

Questions 2, 3 and 4 were tested by firstly exploring the distribution of the data using a Shapiro–Wilk test of normality and was considered to be normally distributed if *p* ≤ 0.05. Normally distributed data between the tasks were assessed using a paired *t*-test and non-normally distributed data were also tested using the Wilcoxon signed-rank test. A difference of the net moment was significant when *p* ≤ 0.05. The results are presented as the absolute and relative mean or mean difference (MD) ± the standard deviation (SDD). All statistical analyses were performed using IBM SPSS Statistics (version 25; IBM Corp., Armonk, NY, USA). In Table 1 can be found a framework to describe per research question the activities.

## 3. Results

Out of the 23 workers who participated in this study, the data of 12 subjects were not useable because of data-acquisition errors in either the IMMUs or sEMG hardware during essential trials for ANN training. In addition, the data of another two subjects performing dynamic and/or heavy tasks contained data-acquisition errors. Therefore, these 14 subjects were excluded from the analysis, leaving a set of data of nine subjects (eight males, one female): four medical disinfectant care workers, three maintenance engineers, and two industrial chemical cleaners. Their mean age was 33.7 ± 10.3 years, height 185 ± 9 cm, and weight 93 ± 12 kg. Eight subjects were right-handed, and one subject was left-handed.

### 3.1. Question 1: Base Quality of ANN-Based Method

The base quality was studied by examining the correlation between the corresponding moment curves from the ANN-based method and the LSM-based method. Table 2 (and Section A.1) shows the correlations. For one subject (subject 6), the data of the calibration movement with 6 kg were not usable and the correlation with the LSM-based model could not be calculated. Therefore, the results of subject 6 were not included in the mean results in Table 2. Hence, this subject was included in the study. The trained ANN method estimation base quality differs from one subject to another.

Good correlations were observed in the trunk bending (mean *r* = 0.89 ± 0.22) and lateroflexion (mean *r* = 0.85 ± 0.22). All subjects but one (subject 8) showed good correlations (*r* ≥ 0.85). Subject 8 showed a weak correlation (*r* ≤ 0.38) for both movements. A moderate correlation was observed during the trunk rotation in the longitudinal axis (*r* = 0.68 ± 0.24) and the anterior–posterior axis (*r* = 0.65 ± 0.24). However, subject 2 showed a weak correlation (*r* = 0.23). Overall, the results were within the acceptable range (*r* > 0.5).

Table 3 compares the RMSE values between the ANN-based method and the LSM-based method. These results indicate a mean estimation error of 9.25 ± 6.01 Nm, relative to the typical peak net moment range from 150 to 220 Nm (see Table 4).

### 3.2. Question 2: Capability to Distinguish Task Intensity

Table 4 shows the tasks per job according to the results of the checklist of physical workload. All subjects, except for one industrial chemical cleaner, perceived the dynamic task as the heaviest task of their job. Small differences in the checklist for physical workload scores were observed, which were related to the diversity in the individual job description.

Table 5 presents the estimated net moments per task together with the experienced workload of the tasks according to the subjects. During all the tasks, the mean net lumbar moment was 25.2 ± 16.8 Nm, the mean peak moment was 179.5 ± 152.9 Nm, and the mean variance was 15.5 ± 11.5 Nm.

Table 6 summarizes the differences between the light and heavy tasks presented for all subjects, with a typical example in Figure 2. The graphs of all subjects can be found in Section A.2. It can be seen that the net moments estimated using the ANN-based method exhibit overall higher moments during heavy tasks with more variances than during light tasks. The differences in the mean net moment between light and heavy tasks were significant (MD = 64.3 ± 72.1%, *p* = 0.028), whereas other differences were not. The perceived workload was significantly higher during the heavy tasks than during light tasks (MD = 5.1 ± 2.1, *p* < 0.001).

### 3.3. Question 3: Capability to Distinguish Static/Dynamic Task Variance

Table 7 summarizes the differences between the static and dynamic tasks presented for all subjects, with a typical example in Figure 3. The graphs of all subjects can be found in Section A.3. It can be seen that the mean net moments of the magnitude estimated using the ANN-based method exhibit overall higher values during dynamic tasks with more variances than during static tasks. The difference in the variance between static and dynamic tasks was significant (MD = 44.8 ± 48.9%, *p* = 0.025). The perceived workload was significantly higher during the dynamic tasks than during static tasks (MD = 2.2 ± 1.5, *p* = 0.002).

### 3.4. Question 4: (a) Symmetrical Lumbar Load

Table 8 summarizes the differences between light and heavy tasks presented while taking into account the direction of the movement around the axis (e.g., flexion vs. extension). Similar results of the intensity levels were observed as shown in Table 6. It was observed that the mean net moment of the magnitude was significantly higher during heavy tasks than during light tasks (MD ≥ 92.3 ± 105.4%, *p* ≤ 0.030) as well as in the singular anterior–posterior axis (MD ≥ 56.8 ± 44.9%, *p* ≤ 0.016). All the other differences were not significant.

Table 9 summarizes the differences between the static and dynamic tasks while taking into account the direction of the movement around the axis. When the direction of the moment around the separated axis was considered, significantly less variance was observed during rotation to the left around the longitudinal axis during dynamic tasks (MD = −90.8 ± 10.5%, *p* = 0.031). In addition, significant differences in the mean moment (||M||, M*_y_*, and M*_z_*) were observed in the positive direction (−47.1% ≤ MD ≤ 302.7%, *p* ≤ 0.042), but were not significant in the negative direction (−54.0% ≤ MD ≤ −276.5%, *p* ≥ 0.326).

## 4. Discussion

The results showed that the ANN-based method can estimate the net moments of the 6 kg test trials with an accuracy of about 9 Nm in comparison with the LSM-based method after being trained with 0 and 10 kg test trials. These results are in line with the research of Baten et al. [19,22], Dolan [25] and Kingma et al. [24] and support the notion that the ANN-based method can be used for evaluating lumbar load exposure patterns and exposure levels in real-life work settings. The feasibility of this approach is also supported by the results of others [42,43], who also used an ANN to predict static postures and net moments driven by static posture data (3D Euler angles) and EMG data and by the results of applying the same method to predict shoulder joint load exposure [34]. The results of the base quality differ per subject due to the individual character of the results. The calibration differs from one subject to another, which can be due to differences between the three calibration sets with three different weights or due to an unidentified event. This resulted in nonperfect calibration for subject 8, with an overall medium to small correlations. This ANN-based method seemed to be capable of distinguishing differences in the intensity level between light and heavy tasks which are in line with the perceived workload. Additionally, it can distinguish the differences in variance level between static and dynamic tasks. When the direction of the moment around the (anterior–posterior, lateroflexion or longitudinal) axis was considered, similar results were observed between light and heavy tasks. However, between static and dynamic tasks, the variance differences of the net moment were not observed. This is because of the differences due to the direction of the moment around the axis. This indicates that it is important to analyze and interpret net moments with different signs separately and thereby acknowledge the (a) symmetrical character of the net moment exertion when analyzing the lumbar load. This study showed that this system can measure the lumbar load in 3D, separately for the direction of the net moment in uncontrolled real-life working conditions.

In studies under more controlled conditions (Kingma et al., 2001, Baten, 2000), a more consistent high estimation accuracy was found, which suggests higher generalizability than the data in this study. The fact that accuracy is less consistent in this study is probably caused by the less controlled experimental conditions and environment, resulting in a lower quality and consistency in the signals recorded and driving the ANN. This is already signaled by the large loss of data due to the technical failures during acquisition. Still, in the end, this study reports the accuracy achieved in spite of the technically challenged acquisition conditions during the actual working conditions at actual work sites.

Like in other (state-of-the-art) research, this study provides insights into the usability and validity of this ANN-based method. This method only requires EMG and IMMU data for monitoring the lumbar load of physically active workers, without requiring any a priori knowledge regarding the load. Tasks were selected on the basis of actual working activities performed in the natural environments of physically demanding jobs. This study also explored the effect of the direction of the moment around the axis with which asymmetrical working routines can be investigated. For example, these could be the case of a paver who uses only one hand to lift tiles or fabric workers who mainly rotate in one direction or axis. Insight into these movements and moments can provide useful information that can help to effectively prevent musculoskeletal complaints. It should also be mentioned that the diversity in the jobs, related workloads, and selected tasks was a challenging aspect, for example, in selecting uniform tasks for the workers because of the differences in their working activities. Real-life tasks are not merely light, heavy, static, or dynamic; rather, a static or dynamic task may also be light or heavy, which results in an overlap between tasks.

The main weakness of this study was the lack of a reference method during actual work. Currently, the mechanical workload of the physically active workers was assessed by observations (video), questionnaires, performance tests, or combinations of motion trackers and force sensors [4,20,44]. Both observations and questionnaires were indirect methods and do not provide information about the working posture or related lumbar load. Reference systems, such as Vicon motion-capture cameras, are not practical or allowed in the real-life working environment [4,27,40,41]. The closest option for a reference method is the method that uses IMMUs and ground reaction forces assessed using an instrumented shoe [20,32,40]. This method, however, has two major drawbacks. The first drawback is that it yields erroneous results every time an external force other than the ground reaction force is exerted on the lower body (e.g., external forces resulting from supporting loads handled with any part of the lower body, or from leaning against a table or workbench, etc.). Another drawback is that it requires wearing heavy and somewhat bulky instrumented shoes, which constitute potential hazards and noncompliance with shoe and work safety functionality and regulations.

The current physical setup is not usable in real-life physically demanding jobs. Gathering data during actual work seemed to be technically challenging and resulted in the data of 14 out of the 23 subjects not being used in the study analysis. According to a power analysis based on the lab study, at least six subjects must be included to validate this model. The remaining nine subjects were sufficient for a first validation in real-life working conditions. However, this large number of dropouts and data processing issues must be prevented. To improve the usability an improved data-acquisition, (hardware) setup is required for future studies and applications. Additionally, further validation in follow-up research with more subjects and in more real-life working situations would increase insight into the quality of the assessment and may help develop insights into how to further improve this method. Preferably, this should be done in selected work situations in which the instrumented shoe method can serve as a reference and/or situations in which all external forces are known in some way. The environment’s influence on the system (e.g., disturbance in the observability of the Earth’s magnetic field) also needs to be further explored [35] and dealt with. Moreover, the design (mechanical) of the system needs to be improved (e.g., by integrating the sensors in the clothes). In addition, the monitored working posture and lumbar load exposure should be linked to ergonomic guidelines to obtain feedback regarding exceeding acceptable loading levels or loading patterns in a way like already pioneered by Baten et al. (e.g., [31,45]). As suggested in these papers, this information can be provided to the user (e.g., using a traffic light model based on EU or NIOSH ergonomic norms, to indicate areas of overload risk). Preferably, the estimating software should do this as fully automated with instant feedback to the user. Such a system would provide workers with feedback regarding their working behaviors and/or work task design and/or working conditions and/or workplace design, which can help improve workers’ behavior, but also work task, organization and conditions and decrease their complaints. Both work postures and their net moments as well as the link to ergonomic guidelines need to, and can, be further explored in follow-up research with the ambulatory methods and tools studied in this paper. Estimated net moment data seem to be very suitable for exposure variance analysis. Adding load exposure pattern analysis to the current analysis of only the amount of load exposure [46] can be useful by means of generating a 2D graph of the joint distribution of intensity of the net moment data dynamically. This exposure variance analysis can be used to link the results to ergonomic guidelines and provide the user with feedback.

The results of this study suggest that not only can the ANN-based method be used in monitoring lumbar back load exposure in physically demanding jobs, but also that it may have the potential to be used in other occupational rehabilitation applications, such as office workers. For office workers, the lumbar load during sedentary behavior could be monitored and investigated, aiming to prevent health problems and physical discomfort related to static working postures. Additionally, this method has the potential to be extended and made usable for full body monitoring or specific areas. Other clinical applications are in the fields of rehabilitation medicine and sports. It may be used individually to assess muscular overload causing or contributing to an (individual) problem and help patients and clinicians to tailor treatments. The ANN-based method could be used in back pain rehabilitation, rehabilitation involving improvement of knee loading behavior, e.g., anterior cruciate ligament (ACL) surgery recovery. Additionally, with this method, injuries could be predicted or prevented in rehabilitation and sports applications and the performance could be improved.

## 5. Conclusions

Lumbar loads could be distinguished with the ANN-based method in terms of intensity and variance levels. The moments in the lumbar region are significantly higher during heavy tasks than during light tasks and the amount of variance is significantly higher during dynamic tasks than during static tasks. The estimated net moments were consistent with the perceived intensity levels and character of the work task in physically demanding occupations. It was concluded that the validity of this sensor-based system was supported, because the differences in the lumbar load were consistent with the perceived intensity levels and character of the work tasks.

## Figures and Tables

**Figure 1 sensors-21-02476-f001:**
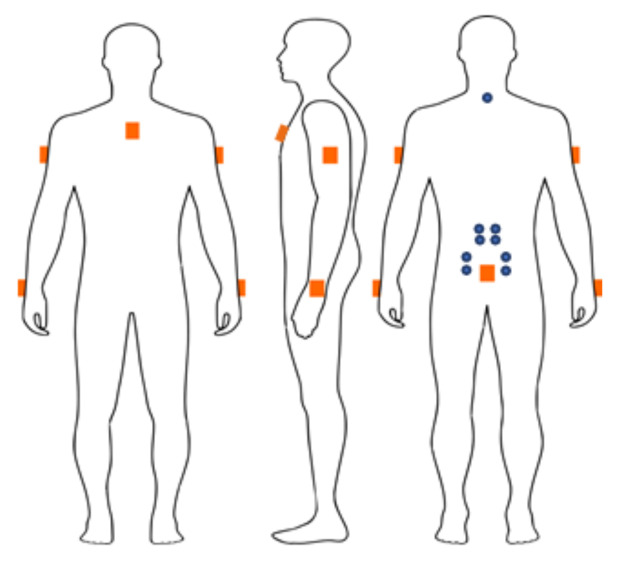
The positioning of the sEMG electrodes and IMMUs on the body. The surface electromyography (sEMG electrodes, blue circles) positioned on the longissimus thoracis muscles at L1 and the iliocostalis lumborum muscles at L2–L3 with a reference electrode placed at the processus spinosus of C7. The inertial magnetic measuring units (IMMUs, orange blocks) positioned on the sternum, upper, and lower (left and right) arms, and pelvis (sacrum) with a front (left), side (middle), and back view (right).

**Figure 2 sensors-21-02476-f002:**
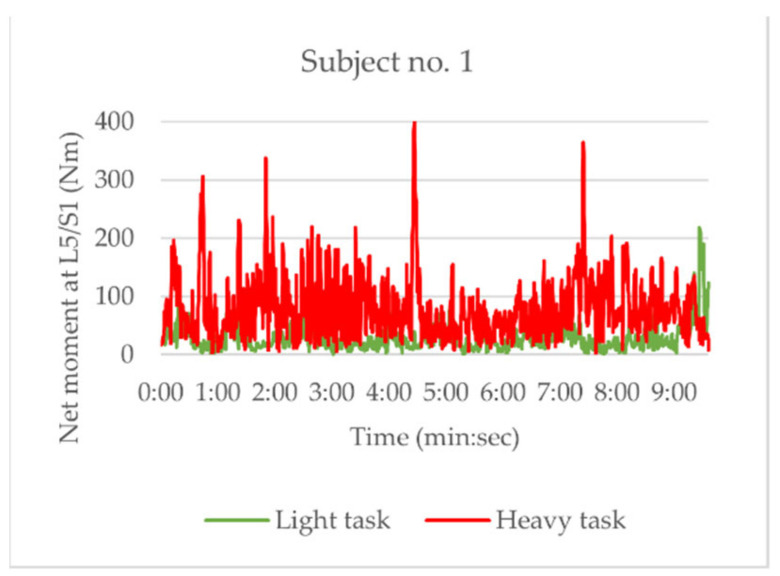
Typical example of net moment curves during light (green) and heavy (red) tasks (Subject 1).

**Figure 3 sensors-21-02476-f003:**
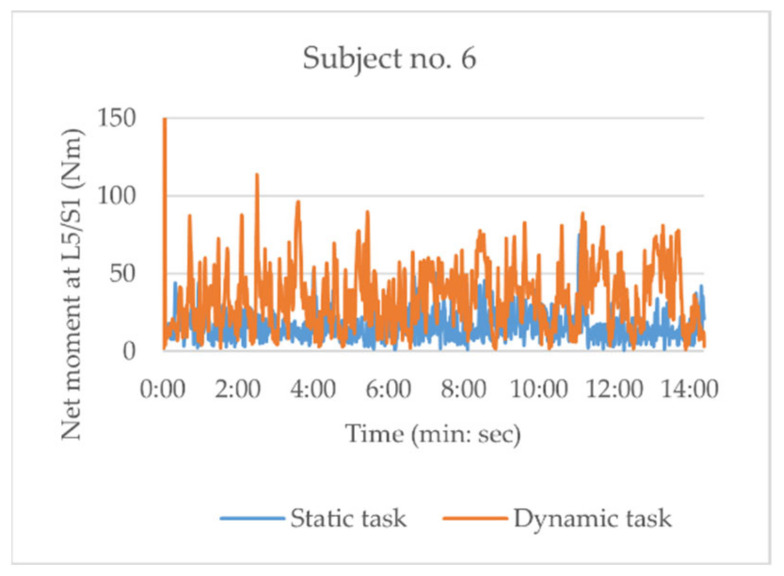
Typical example of net moment curves during static (blue) and dynamic (orange) tasks (Subject 6).

**Table 1 sensors-21-02476-t001:** Framework study design. A framework to describe per research question the activities in terms of inertial magnetic measurement units (IMMUs) and surface electromyography sensors (sEMG)), discriminant validity and related criteria with artificial neural network (ANN) and linked segment model (LSM).

Step	Research Question	Activity	Discriminant Validity	Criteria
1	1	Calibration measurement with IMMUs and sEMG	ANN vs. LSM method	Mean *r*^2^ of subject ≥ 0.5
2	2, 4	Physical workload checklist	Light vs. heavy task	Light: lowest workload per job Heavy: highest workload per job
	3, 4		Static vs. dynamic task	Static: same posture at least 4 s Dynamic: different working postures
3	2, 3, 4	Performing work-related tasks with IMMUs and sEMG	Test research questions	*r* ≥ 0.5 *p* ≤ 0.005
4	2, 3, 4	Perceived workload questionnaire	Test research questions	*p* ≤ 0.005
5	1, 2, 3, 4	Calibration measurement with IMMUs and sEMG	Check measurement quality	N/A

**Table 2 sensors-21-02476-t002:** ANN-based method performance in handling known loads. Shown are the correlations between the net moment at L5/S1 estimated with the ANN-based method and with the LSM-based method. Correlation is represented by the Pearson correlation coefficient (*r*) and determination coefficient (*r*^2^) are shown only for the axis of movement in each task (i.e., in the mediolateral axis (*y*) for the trunk bending tasks, in the anterior–posterior axis (*x*) for the lateroflexion tasks, and the longitudinal axis (*z*) for the rotation tasks, respectively. Shown are individual values for each subject plus the mean and standard deviation (SDD) over all subjects. No valid data were obtained for subject number 6 for reasons of partially missing data in the 6 kg trial.

			Subject										
Movement	Axis		1	2	3	4	5	6	7	8	9	Mean	SDD
Bending	*y*	*r*	0.97	0.98	0.96	0.97	0.98	-	0.96	0.35	0.98	0.89	0.22
		*r* ^2^	0.94	0.95	0.93	0.93	0.96	-	0.92	0.12	0.96	0.84	0.29
Latero-flexion	*x*	*r*	0.94	0.93	0.95	0.95	0.94	-	0.85	0.32	0.92	0.85	0.22
		*r* ^2^	0.89	0.87	0.91	0.90	0.88	-	0.72	0.10	0.84	0.76	0.27
Rotation	*z*	*r*	0.63	0.23	0.84	0.84	0.81	-	0.76	0.38	0.92	0.68	0.24
		*r* ^2^	0.40	0.06	0.70	0.70	0.65	-	0.58	0.15	0.84	0.51	0.28

**Table 3 sensors-21-02476-t003:** RMSE between the ANN-based method and the biomechanical model. The root means square error (RMSE) between the net moment curves at L5/S1 was estimated by the ANN-based method and the LSM-based method. Values are shown for the three movements trunk bending, lateroflexion and rotation with the anterior–posterior axis (*x*), mediolateral axis (*y*) and longitudinal axis (*z*). Shown are individual values for each subject plus the mean and standard deviation (SDD) over all subjects. No valid data were obtained for subject number 6 for reasons of partially missing data in the 6 kg trial.

		Subject										
Movement	Axis	1	2	3	4	5	6	7	8	9	Mean	SDD
Bending	*x*	12.5	5.00	10.7	4.49	7.09	-	5.24	17.2	4.84	8.38	4.63
	*y*	8.95	8.65	9.85	8.56	10.4	-	21.1	9.31	6.17	10.4	4.51
	*z*	6.49	3.91	5.66	3.15	5.87	-	6.70	9.64	2.61	5.50	2.27
Lateroflexion	*x*	17.2	10.9	12.5	7.12	13.4	-	19.5	31.3	11.9	15.5	7.44
	*y*	23.7	11.2	11.7	6.83	16.6	-	13.2	19.5	11.7	14.3	5.37
	*z*	11.3	8.88	10.4	3.48	9.83	-	8.73	24.0	6.92	10.4	5.98
Rotation	*x*	2.98	2.55	3.27	3.89	6.07	-	5.15	18.9	2.73	5.68	5.47
	*y*	4.28	4.10	3.73	4.06	6.26	-	6.05	21.5	4.33	6.79	6.02
	*z*	5.01	5.73	4.41	4.12	6.32	-	6.90	12.8	4.87	6.27	2.80

**Table 4 sensors-21-02476-t004:** Tasks per job type. Based on the results of the checklist for the physical workload.

Task Perception	Medical Disinfect Care Worker	Maintenance Engineer	Industrial Chemical Cleaner
Light	Changing personal protective working clothing before or after working in contaminated space	Administration of technical maintenance service	Disassemble parts of a gas mask as preparation for cleaning
Static	Assembly or lamination of surgical instruments	Tinkering under a machine to fix or loosen components	Cleaning chemical hazard suit in sink
Heavy and dynamic	Carrying bins of 3 up to 10 kg over a distance of about 1 m	Moving (pushing and/or pulling) bin with wastewater of 1000 kg or carrying a toolbox of 35 kg over a distance of about 50 m	Carrying bins of 5–10 kg over a distance of 50 m

**Table 5 sensors-21-02476-t005:** Net moments, load ranking, and perceived workload for each task. The questionnaire workload factor with 1 = light work and 5 = very heavy task. The experienced workload (Borg CR-10) according to the subjects with 0 = not burdensome and 10 = extremely heavy.

	Net Moment (Nm)			Questionnaire Load Factor (1–5)	Perceived Workload (Borg 0–10)
Task Perception	Mean	Peak	Variance		
Light task	18.7 ± 8.1	166.4 ± 195.5	13.0 ± 10.6	1.0 ± 0.0	0.9 ± 0.8
Static task	26.3 ± 19.2	153.5 ± 106.1	13.7 ± 9.9	3.6 ± 1.3	3.8 ± 1.6
Heavy and dynamic task	30.7 ± 20.0	218.5 ± 154.4	19.8 ± 13.8	4.8 ± 0.4	6.0 ± 2.0

**Table 6 sensors-21-02476-t006:** Light vs. heavy tasks. Shown are the absolute (Nm) and relative (%) differences between the light tasks and the heavy tasks through mean difference (MD: heavy–light task) and its standard deviation (SDD) plus 95% confidence interval (CI) of these differences for the moment magnitude (||M||) with *p*-value of paired *t*-test.

	Absolute (Nm)		Relative (%)		
Parameter	MD ± SDD	(95% CI)	MD ± SDD	(95% CI)	*p*
Mean	12.0 ± 13.5	[1.7;22.4]	64.3 ± 13.5	[8.9;119.8]	0.028
Peak	52.1 ± 256.9	[−145.4;249.6]	23.9 ± 117.6	[−66.5;114.2]	0.560
Variance	6.8 ± 9.4	[−0.4;14.0]	52.1 ± 71.8	[−3.1;107.3]	0.061

**Table 7 sensors-21-02476-t007:** Static vs. dynamic tasks. Shown are the absolute (Nm) and relative (%) differences between the static tasks and the dynamic tasks through mean difference (MD: dynamic-static task) and its standard deviation (SDD) plus 95% confidence interval (CI) of these differences for the moment magnitude (||M||) with *p*-value of paired *t*-test.

	Absolute (Nm)		Relative (%)		
Parameter	MD ± SDD	(95% CI)	MD ± SDD	(95% CI)	*p*
Mean	4.42 ± 8.03	[−1.8;10.6]	16.8 ± 30.6	[−6.7;40.4]	0.137
Peak	65.0 ± 138.6	[−41.5;171.5]	42.3 ± 90.3	[−27.0;111.7]	0.197
Variance	6.13 ± 6.69	[1.0;11.3]	44.8 ± 48.9	[7.2;82.4]	0.025

**Table 8 sensors-21-02476-t008:** Direction of the net moment of the magnitude and around the axis of the light vs. heavy tasks. Shown are the absolute (Nm) and relative (%) differences between the light tasks and the heavy tasks through mean (MD: heavy–light task), standard deviation (SDD), and 95% confidence interval (CI) of these differences for the anterior–posterior (Mx) with lateroflexion to the left (positive) and right (negative), mediolateral (My) with flexion (positive) and extension (negative), longitudinal (Mz) with rotation to the left (positive) and right (negative) axes separately and for the moment magnitude (||M||) with a *p*-value of paired *t*-test.

			Absolute (Nm)		Relative (%)		
Parameter	Direction	Axis	MD ± SDD	(95% CI)	MD ± SDD	(95% CI)	*p*
Mean	Positive	||M||	152.0 ± 92.8	[80.6;223.4]	1324.0 ±808.6	[702.4;1945.5]	0.001
		Mx	5.6 ± 4.4	[2.2;8.9]	56.8 ± 44.9	[22.3;91.3]	0.005
		My	−0.6 ± 8.6	[−7.2;6.0]	−2.5 ± 33.6	[−28.4;23.3]	0.828
		Mz	1.3 ± 3.6	[−1.5;4.1]	15.6 ± 43.7	[−18.0;49.1]	0.317
	Negative	||M||	9.5 ± 10.9	[1.8;17.9]	92.3 ± 105.4	[11.3;173.3]	0.030
		Mx	8.4 ± 8.3	[2.1;14.8]	95.0 ± 93.6	[23.1;166.91]	0.016
		My	3.4 ± 13.5	[−5.7;12.5]	24.4 ± 97.2	[−41.0;90.4]	0.424
		Mz	4.8 ± 7.2	[−0.7;10.3]	53.5 ± 79.7	[−7.8;114.8]	0.079
Variance	Positive	||M||	6.6 ± 13.5	[−2.5;15.7]	37.3 ± 76.8	[−14.2;88.9]	0.138
		Mx	1.3 ± 7.5	[−3.8;6.4]	11.9 ± 68.0	[−33.8;57.5]	0.575
		My	1.1 ± 11.2	[−6.5;8.6]	6.8 ± 72.4	[−41.8;55.5]	0.762
		Mz	−1.1 ± 6.7	[−5.6;3.5]	−12.2 ± 76.7	[−63.7;39.3]	0.610
	Negative	||M||	2.2 ± 9.5	[−4.2;8.6]	17.3 ± 73.7	[−32.2;66.8]	0.453
		Mx	5.3 ± 9.6	[−1.2;11.8]	38.1 ± 69.4	[−8.6;84.7]	0.099
		My	1.1 ± 8.6	[−4.7;6.8]	9.9 ± 79.0	[−43.1;63.0]	0.686
		Mz	2.3 ± 8.4	[−3.3;8.0]	24.3 ± 88.4	[−35.0;83.7]	0.382

**Table 9 sensors-21-02476-t009:** Direction of the net moment of magnitude and around the axis of the static vs. dynamic tasks. Shown are the absolute (Nm) and relative (%) differences between the static tasks and the dynamic tasks through mean difference (MD: static–dynamic) and its standard deviation (SDD) plus 95% confidence interval (CI). These are shown for the net moments around each of the three axes separately (in the right-handed axes frame): for the moments around the anterior–posterior axis (Mx), with the net moment positive for direction of rotation to the left; for the moments around the mediolateral axis (My), with the net moment positive in the forward flexion direction; for the moments around the longitudinal axis (Mz), with the net moment positive in the direction of rotation to the left and for the moment magnitude (||M||) with a *p*-value of paired *t*-test.

			Absolute (Nm)		Relative (%)		
Parameter	Direction	Axis	MD ± SDD	(95% CI)	MD ± SDD	(95% CI)	*p*
Mean	Positive	||M||	100.5 ± 132.8	[11.3;189.7]	302.7 ± 399.7	[34.1;571.2]	0.031
		Mx	−6.6 ± 14.8	[−16.6;3.3]	−34.5 ± 76.9	[−86.1;17.1]	0.168
		My	−18.1 ± 25.7	[−35.4;−0.8]	−47.1 ± 66.9	[−92.0;−2.1]	0.042
		Mz	−6.5 ± 8.6	[−12.3;−0.7]	−45.2 ± 60.2	[−85.7;−4.8]	0.032
	Negative	||M||	−44.9 ± 156.2	[−149.9;60.0]	−276.5 ± 960.7	[−921.9;368.9]	0.362
		Mx	−7.7 ± 38.6	[−33.6;18.3]	−54.0 ± 272.6	[−237.1;129.1]	0.526
		My	−22.4 ± 78.0	[−74.8;30.0]	−161.7 ± 561.8	[−539.1;215.7]	0.362
		Mz	−6.7 ± 21.4	[−21.0;7.7]	−59.0 ± 189.5	[−186.3;68.3]	0.326
Variance	Positive	||M||	−6.9 ± 44.9	[−37.1;23.2]	−39.4 ± 254.7	[−210.5;131.7]	0.619
		Mx	–5.1 ± 12.5	[−13.6;3.3]	−46.2 ± 112.8	[−122.0;29.6]	0.204
		My	−4.8 ± 13.1	[−13.6;3.9]	−31.2 ± 84.3	[−87.8;25.4]	0.248
		Mz	−8.0 ± 10.5	[12.1;0.9]	−90.8 ± 10.5	[−171.2;10.4]	0.031
	Negative	||M||	–27.2 ± 97.5	[−92.7;38.3]	−210.5 ± 755.0	[−717.7;296.8]	0.377
		Mx	−1.5 ± 25.3	[−18.5;15.5]	−10.7 ± 182.2	[−133.1;111.7]	0.849
		My	−6.4 ± 26.1	[−23.9;11.1]	−59.0 ± 240.5	[−220.6;102.5]	0.434
		Mz	−3.3 ± 17.6	[−15.1;8.5]	−34.4 ± 183.7	[−157.8;89.1]	0.549

## Data Availability

The data presented in this study are available on request from the corresponding author. The data are not publicly available due to it possibly containing information that could compromise the privacy of some research participants.

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
