# Peer review of "Automatically Determining Lumbar Load during Physically Demanding Work: A Validation Study"

_sensors, 2021, doi:10.3390/s21072476_

Round 1

Reviewer 1 Report

The main purpose of this paper is to validate an artificial neural network (ANN) based method for heavy/light, static/dynamic lumbar loading on the spinal L5/S1 in actual, physically demanding work situations. The paper explores the emerging use of ANN with EMG and kinematic inputs to discriminate light/heavy and static/dynamic tasks. However, more details on the ANN model is needed.

  • Since the ANN method is crucial to this paper theme, more methodological details on the model would help support the paper claims, such as the number of iterations to complete training and the criterion to stop training? How was overtraining avoided?
  • What features of EMG were used to feed into the ANN model? How sensitive is the model to noise?
  • Ln 247-248, since the trained ANN differs in base quality from one subject to another, can the authors discuss the generalizability of the ANN model?
  • The role of Subject 6 is unclear, can you clarify in which experiments Subject 6’s data was included?
  • In table 1, can the authors discuss the lack of correlation for rotation in subject 2?

Minor comments

  1. Some spelling errors; Ln 134 ‘alle’
  2. Ln 210-211 do the authors refer to question 2 instead of question 3?

Reviewer 2 Report

The article aims at testing an artificial neural network-based method for assessing the work load, considering experimental data collected by IMUs and sEMG sensors.

The study is well written and supported by a real case study.

However, there are some issues and doubts, which are described as follows.

I would recommend the authors to make revisions accordingly in order to improve this manuscript. Before considering this paper ready for publication, revisions are mandatory.

SECTION 1

1) At lines 41-42 the author stated that motion capture systems are useful for monitoring the lumbar load... Actually these systems provide only kinematic data; 

2) the literature review is not fully comprehensive and in some parts it is explained in an approximate manner;

3) there are several researches that analysed data collected by a combined use of IMUs and sEMG in real working environment, (such as Monaco et al. Biomechanical overload evaluation in manufacturing: A novel approach with sEMG and inertial motion capture system integration. Advances in Intelligent Systems and Computing Volume 818, 2019, Pages 719-726; Ranavolo et al. Wearable monitoring devices for biomechanical risk assessment at work: current status and futur challenges - a systematic review. International Journal of Environmental Research and Public health Volume 15, 2018; etc.). Please, improve the literature review.

SECTION 2

4) Please, detail some more information about the used checklist and list the questions (section 2.2);

5) A schematic framework of the adopted procedure and the related data analysis step would help to make the setting of your work clearer, as well as a more fluent reading; 

6) How did you synchronize the IMU and EMG systems?

SECTION 3

7) Table 3 is not clear. 

8) Why did you consider data from 9 subjects sufficient to validate your model? Please, justify this aspect well.

Reviewer 3 Report

Major comments

  1. Rationale and background information would benefit from stronger, more appropriate, or additional relevant supporting literature. The text would also benefit from additional discussion of how this work fits in the context of relevant work and findings from other groups.
    • Some references appear to be presentations without a written/recorded component (12, 19, 27, 32, 35, 42) and one is an unpublished manuscript (33). Since readers cannot review these references, they do not add support to any of the statements for which they are cited.
    • While referencing conference proceedings is acceptable in the absence of other references or as part of a larger list amongst article or book references, statements are weakened if they rely upon conference proceedings alone. Can any of these references (18, 20, 28, 29, 30) be replaced or supported with journal article or book references?
    • Seemingly relevant literature about lumbar loading estimation with ANNs is not discussed, for example, work by Arjmand and colleagues such as:
      • Gholipour, N. Arjmand. Artificial neural networks to predict 3D spinal posture in reaching and lifting activities; Applications in biomechanical models. Journal of Biomechanics. Volume 49, Issue 13, 2016, Pages 2946-2952.
      • Aghazadeh, N. Arjmand, A.M. Nasrabadi. Coupled artificial neural networks to estimate 3D whole-body posture, lumbosacral moments, and spinal loads during load-handling activities. Journal of Biomechanics. Volume 102, 2020, Article 109332.
    • Please verify that references cited support claims. For example:
      • Line 74 cites reference 23, Valevicus et al., 2018, but that review does not discuss ANN-based methods.
      • Line 194 cites reference 39, Kristiansen et al., 2019, but the intervals and nomenclature stated in reference 39 do not match those of the statement for which it is cited.
  1. Additional methodological detail is needed.
    • It is unclear why an LSM-based method is chosen as the “gold standard” against which the ANN-based method is evaluated. Please provide rationale for this decision.
    • Insufficient detail is provided to repeat the LSM-based methods. The first reference cited (21) describes use of marker-based motion capture and forceplates for the LSM method while reference 27 has no print component cited. Please add a summary of the LSM method (including number and type of marker-based system and forceplates used) and update the text and references as needed so that the reader may refer to them for additional methodological detail.
    • Similarly, insufficient detail is provided to reproduce the ANN method.
      • Please describe what data were used by the neural network (e.g., which body segment and joint kinematics; were angles, angular velocity, segment velocities, and/or other information used?).
      • Please clarify whether EMG was collected bilaterally.
      • Please summarize the optimal settings (line 126) for the ANN-based method.
    • Please include the questionnaire referred to in line 137.

Minor comments

  1. Please comment on why subject 8 displayed such poor correlation between the ANN and LSM-based methods. Was the data from this subject used in subsequent analyses?
  2. Consider providing the correlation information for the additional axes that were left out of Table 1 (such as the “anterior-posterior axis” referenced in line 262) in a supplementary table. Consider also providing in the supplementary material example figures with the ANN predictions compared to the LSM-method similar to comparisons shown in Figures 2 and 3.
  3. Line 266 states “…typical net moment ranges from 150 to 220 Nm…” but when compared to the data presented in Table 4, this appears to be the range of the peak net moment, not the net moment range. Please clarify.
  4. Please add a legend to the plots in Figures 2 and 3.
  5. In Tables 5-8, please clarify what test the reported p-values refer to and which quantities were compared.
  6. Line 354: Reporting the average instead of the range or maximum difference is a bit misleading. Consider rephrasing this statement.
  7. The statements in lines 364-367 advocate analyzing moments of different signs separately, which would eliminate the differences observed between the static and dynamic tasks using the ANN-based method. This change in the analysis would contradict the finding that variance is higher during dynamic tasks compared to static tasks. Please clarify this point.
  8. Correction of various typographical errors and changes in word choice would improve the clarity of the manuscript.
    • To avoid confusion, consider using “height of the subject” instead of “length of the subject” (e.g., lines 64, 241).
    • Use the same wording to describe the job types of the participants in the text and in Table 3.
    • Standardize results terminology throughout the results section text and table captions. For example, Line 325 “mean net moment” compared to the Table 7 caption in line 333 “moment magnitude”.
    • Consider in line 341 separating the positive and negative mean differences reported or reporting the magnitude of the mean difference in the text.
    • Please address various typographical and grammatical errors. For example:
      • Line 245: “…moment curves form ANN based method en the LSM based method.”
      • Caption for Table 1, line 253: “…for the lateroflexion tasks (y)…”
      • Caption for Table 2, line 270: “Values are shown only for the axis of movement in each task…”
      • Caption for Table 2, line 272: “…for the lateroflexion axis (y)…”
      • Lines 415-416 there are two “e.g.,” in a row with no closing parenthesis.
      • Lines 425-427 is an incomplete sentence.
    • Please clearly show how the results of this study suggest that ANN-based methods would be useful in the other applications and fields listed in lines 428-434.
    • Please clarify what is meant by “in-vivo setup” in line 436.

Round 2

Reviewer 2 Report

The authors significantly improved their article by considering the suggestions from the first review. In addition, they provided comprehensive responses to clarifications requested by the reviewer.

I consider the article suitable for publication.

Author Response

We thank the reviewer for this positive comment.